# A Three-Dimensional Inversion Method for Small-Scale Magnetic Objects Based on Normalized Magnetic Source Strength

**DOI:** 10.3390/mi13111813

**Published:** 2022-10-24

**Authors:** Ziming Cai, Zhining Li, Hongbo Fan, Qingzhu Li, Bo Wang

**Affiliations:** Department of Vehicle and Electrical Engineering, Shijiazhuang Branch, Army Engineering University of PLA, Shijiazhuang 050003, China

**Keywords:** shape inversion, magnetic target, normalized magnetic source strength, small scale target

## Abstract

The exploration of some dangerous or important small-scale magnetic objects requires accurate three-dimensional inversion results. In this paper, a three-dimensional inversion method for small-scale magnetic objects is proposed. Normalized magnetic source strength, which is weakly sensitive to the magnetization direction, is used for inversion, which avoids the influence of the remanence of magnetic objects. The planted inversion method is improved to make the inversion results more similar to the real results; normalized magnetic source strength is used to estimate the center position of the magnetic objects, which provides a priori information for the inversion; the weighting function is added in the inversion process to improve the inversion accuracy. The simulation and experimental results show that the method is not affected by the remanence, and effectively reduces the interference of non-target field sources. The obtained inversion results have higher accuracy.

## 1. Introduction

In the process of detecting some important or dangerous small magnetic targets (such as unexploded ordnance, Ferromagnetic underground pipes, etc.), we hope to obtain the shape and position of the target, so as to guide the follow-up unexploded ordnance removal and other work, and reduce the danger or loss caused by the removal of unexploded ordnance, etc. Therefore, it is necessary to perform 3D inversion of these magnetic targets by using magnetic field data [1,2]. There has been a lot of research in the field of geological exploration using gravity fields for 3D inversion, and these methods are also suitable for 3D inversion of the magnetic field after certain modifications. With the development of the magnetic gradient tensor system, we can obtain more informative magnetic gradient tensor data, making it possible to use magnetic field data for 3D inversion [3,4,5]. The method of magnetic field detection cannot detect non-magnetic objects, but it is very effective for magnetic targets such as unexploded bombs.

At present, the mainstream 3D inversion algorithms can be divided into two categories [6]. (1) One is to obtain the three-dimensional shape of the magnetic object by calculating the distribution of the field source parameter values [7,8]. For example, the magnetization distribution in the area to be measured is calculated, and a threshold is set to determine a boundary of the magnetization distribution, which is used as the inversion result. Such methods require solving underdetermined equations. To solve the underdetermined equation, the regularization method proposed by Tikhonov [9] is generally used. The inversion process needs to add constraints to obtain accurate results, and requires a great deal of computation to optimize the regularization parameters. The calculation is difficult, and the boundaries of the inversion results are not clear. (2) Another method does not solve the underdetermined equation, but assigns field source parameters to the geometry, such as magnetization, magnetization direction, etc., and fits the shape of the magnetic object through the change of the geometry [10]. Krahenbuhl and Li [11] used a combination of simulated annealing and genetic algorithms to estimate the density distribution of salt bodies. This method calculates the geometry shape simply and directly, but the amount of calculation is large, and the inversion result is not accurate. Uieda et al. [12,13] proposed a method for inversion using planting density anomalies. This method effectively reduces the amount of calculation and improves the accuracy of the inversion results. However, this method needs to know certain a priori conditions, such as the approximate location of the target, etc., but the article does not give a method for obtaining the relevant prior conditions. In addition, this method is aimed at the gravitational field, so it does not consider the magnetization direction in the magnetic field. Shearer’s [14] analysis of the effect of remanence shows that when the inversion magnetization direction differs from the real magnetization direction by more than 15°, a completely wrong inversion result may be generated. This effect is especially pronounced for small-scale targets. Therefore, Li et al. [15] used magnetic anomaly modulus data, which are less affected by the magnetization direction for inversion, but the data inversion ability is poor, and they are still affected to a certain extent when the remanence is strong.

In order to solve the shortcomings of existing methods, in this paper, a method for the 3D inversion of small-scale magnetic objects using normalized magnetic source strength is proposed. We used the normalized magnetic source strength derived from the magnetic gradient tensor for inversion. These magnetic field data are weakly sensitive to the direction of magnetization. Rehman and Abdelazeem et al. identified the practical application capabilities of these data [16]. Yin et al. used the normalized magnetic source strength to estimate the position and magnetic moment of the magnetic dipole [17]. Zhou et al. used normalized magnetic source strength and gravity data for joint inversion [18]. Zhang et al. estimated the magnetization direction by using the normalized magnetic source strength [19]. These applications illustrate the practicability of normalized magnetic source strength. Before the inversion, we use the normalized magnetic source strength to estimate the center position of the magnetic source and use this as an a priori condition for inversion. Based on the planting inversion method of Uieda et al., the scale factor and cross-correlation coefficient are added, and this gives the inversion results a higher overall similarity to the actual results. Considering that the normalized magnetic source strength decays rapidly with depth and is easily interfered with by non-target magnetic sources, we added a depth weighting function and a horizontal boundary weighting function in the inversion process. Finally, our algorithm is verified with simulation data and actual measurement data. Compared with the existing methods, this method does not need to estimate the magnetization of magnetic objects and can be used for inversion in the case of multiple magnetic sources. The inversion results have higher similarity with the actual shape, that is, they have higher inversion accuracy.

## 2. Methods

### 2.1. Normalized Magnetic Source Strength Definition and Magnetic Source Location Estimation

The magnetic gradient tensor is defined as the spatial variation rate of the three components of the magnetic field vector in the orthogonal direction, that is, the partial derivatives of the magnetic field component are obtained on the three axes. The relationship between the components of the magnetic field vector and the magnetic gradient tensor is shown in Figure 1.

According to Figure 1, the components of the magnetic field vector along the x, y, and z directions are *Bx*, *By*, and *Bz*. The change rates of *Bx* in the x, y, and z directions are *Bxx*, *Bxy*, and *Bxz*. The change rates of *By* in the x, y, and z directions are *Byx*, *Byy*, and *Byz*. The change rates of *Bz* in x, y, and z directions are *Bzx*, *Bzy*, and *Bzz*.

We convert the above data volumes into matrix representations. If *B* represents the magnetic field vector, the magnetic gradient tensor matrix ***G*** can be expressed as:(1)$$G=\left[\begin{array}{c} \frac{\partial }{\partial x}\\ \frac{\partial }{\partial y}\\ \frac{\partial }{\partial z} \end{array}\right]\left[\begin{array}{ccc} B_{x} & B_{y} & B_{z} \end{array}\right]=\left[\begin{array}{ccc} B_{xx} & B_{xy} & B_{xz}\\ B_{yx} & B_{yy} & B_{yz}\\ B_{zx} & B_{zy} & B_{zz} \end{array}\right]$$

Diagonalizing the tensor matrix *G*:(2)$$G={\left[\begin{array}{ccc} v_{1} & v_{2} & v_{3} \end{array}\right]}^{T}\left[\begin{array}{ccc} \lambda_{1} & & \\ & \lambda_{2} & \\ & & \lambda_{3} \end{array}\right]\left[\begin{array}{ccc} v_{1} & v_{2} & v_{3} \end{array}\right]$$
where $\lambda_{1}$,$\lambda_{2}$, and $\lambda_{3}$ are the eigenvalues and $\lambda_{1}$≥$\lambda_{2}$≥$\lambda_{3}$, $v_{1}$,$v_{2}$,$v_{3}$ are the corresponding eigenvectors. This means that at any measurement point it is possible to find a Cartesian system in which the gradient tensor has zero off-diagonal terms. Therefore, under any coordinate transformation, the tensor matrix has three quantities that are not transformed with the coordinates [20]:(3)$$\left\{\begin{array}{l} I_{0}=trace(G)=\sum_{i=1}^{3}B_{ii}=0\\ \begin{array}{c} I_{1}=B_{xx}B_{yy}+B_{yy}B_{zz}+B_{xx}B_{zz}-B_{xy}^{2}-B_{yz}^{2}-B_{xz}^{2}\\ I_{2}=\det(G)=B_{xx}\left(B_{yy}B_{zz}-B_{yz}^{2}\right)+B_{xy}(B_{yz}B_{xz}-B_{xy}B_{zz})\\ \;+B_{xz}\left(B_{xy}B_{yz}-B_{xz}B_{yy}\right) \end{array} \end{array}\right.$$

The eigenvalues here satisfy the characteristic equation [20]:(4)$$\lambda_{i}{}^{3}-I_{0}\lambda_{i}{}^{2}+I_{1}\lambda_{i}-I_{2}=\lambda_{i}{}^{3}+I_{1}\lambda_{i}-I_{2}=0$$
where i = 1, 2 or 3.

According to the tensor invariants, the following settings are given [20]:(5)$$\left\{\begin{array}{l} C=\sqrt[{3}]{\frac{I_{2}}{2}+\sqrt{{\left(\frac{I_{2}}{2}\right)}^{2}+{\left(\frac{I_{1}}{3}\right)}^{3}}}\\ D=\sqrt[{3}]{\frac{I_{2}}{2}-\sqrt{{\left(\frac{I_{2}}{2}\right)}^{2}+{\left(\frac{I_{1}}{3}\right)}^{3}}} \end{array}\right.$$

The expression of the normalized magnetic source strength NSS given by Beiki [21] is:(6)$$\left\{\begin{array}{l} \begin{array}{l} \lambda_{1}=C+D\\ \lambda_{2}=-\frac{C+D}{2}-\frac{C-D}{2}\sqrt{-3}\\ \lambda_{3}=-\frac{C+D}{2}+\frac{C-D}{2}\sqrt{-3} \end{array}\\ NSS=\sqrt{-\lambda_{2}{}^{2}-\lambda_{1}\lambda_{3}}=\frac{\mu }{{\left|\vec{r}-{\vec{r}}_{0}\right|}^{4}} \end{array}\right.$$
where μ is the physical quantity related to the magnetization. $\vec{r}$ and ${\vec{r}}_{0}$ are the position vectors of the observation point and the field source point. The specific derivation process can be found in the literature [21]. Through the calculation process and expression of the normalized magnetic source strength, it can be found that these magnetic field data are derived from the eigenvalues of the magnetic gradient tensor matrix, and there are no parameters related to the magnetization direction in its expression. Therefore, the normalized magnetic source strength is weakly sensitive data in the direction of magnetization. Inversion using these data can ignore the influence of the remanence of the magnetic object, so it is not necessary to estimate the magnetization direction of the magnetic object in the inversion process. The inversion module can be assigned values using the background magnetic field direction. In the inversion process, the position of the initial model needs to be determined, and the position of the initial model is best set at the center of the magnetic object. If the position of the initial model has a large deviation from the position of the magnetic object, the error of the inversion result will be greatly increased. For small-scale magnetic objects, it is not advisable to choose the approximate location by experience. Because the scale is small, the selected position is prone to large deviations, so it is necessary to estimate the center position of the magnetic object.

According to Equation (4), the maximum value of the normalized magnetic source strength of the magnetic dipole is at the observation point just above it, and this phenomenon is not affected by the magnetization direction. Therefore, the local maximum of the normalized magnetic source strength in the region can be taken as the horizontal coordinate of the magnetic object. However, when the detection distance is close to the magnetic object, the normalized magnetic source strength will have multiple extreme points. When the detection distance is much larger than the scale of the magnetic source, the magnetic source can be regarded as a magnetic dipole [22]. In this case, the maximum value point can be used to judge the horizontal position of the magnetic object. Therefore, a small-scale magnetic object only needs to extend its observation data upward for a certain distance and can be regarded as a magnetic dipole. The continuation distance is the distance when the continuation data happen to have a unique maximum value. In the actual measurement process, we can only obtain the measurement data of a certain height, so we use the method of frequency domain conversion and use the Fourier transform to obtain the normalized magnetic source at different heights. Taking the coordinates of the unique maximum point in the area in the continuation data as the horizontal position coordinates of the magnetic object, the formula for upward continuation is:(7)$$NSS(x,y,h)=F^{-1}[e^{-\sqrt{k_{x}{}^{2}+k_{y}{}^{2}}h}nss(k_{x},k_{y},0)]$$

In Equation (7), $F^{-1}$ represents the inverse Fourier change, $nss(k_{x},k_{y},0)$ is the frequency domain expression of the measured data, $k_{x}$ and $k_{y}$ are the wavenumbers in the x and y directions respectively, and *h* is the upward continuation distance.

Once the horizontal position is determined, its depth can be estimated using the cross-correlation of the equivalent sources. We set a series of equivalent magnetic source models at equal intervals at the position just below the horizontal coordinate. We then calculate the cross-correlation between the forward data of the equivalent magnetic source model at different depths and the measured data and compare the cross-correlation coefficients at different depths. The depth with the largest cross-correlation coefficient is the best estimated depth of the magnetic object. When the shape of the magnetic objects is known, the size of the equivalent magnetic source can be set to the same proportion as the magnetic object. Magnetic dipoles can also be used instead when the shape of the magnetic object is unknown. The equivalent source cross-correlation method makes full use of the advantages of grid data and applies enough data to the localization process. Furthermore, the cross-correlation method has good tolerance to noise, so this method can obtain better depth estimation results in the actual measurement environment with noise.

The calculation formula of the cross-correlation coefficient between the forward modeling data of the equivalent source and the actual measured data is:(8)$$C=\frac{\sum_{i=1}^{N}\left[(NSS_{f}(i)-\bar{NSS_{f}})\times (NSS(i)-\bar{NSS}\right)]}{\sqrt{\sum_{i=1}^{N}\left[{\left(NSS_{f}(i)-\bar{NSS_{f}}\right)}^{2}\times {\left(NSS(i)-\bar{NSS}\right)}^{2}\right]}}$$
where *N* is the number of measurement points, $NSS_{f}$ is the forward value of the equivalent magnetic source, $\bar{NSS_{f}}$ is the mean value of the forward value of the equivalent magnetic source, NSS is the measured data, and $\bar{NSS}$ is the mean value of the measured data.

In order to prove the practicability of the method proposed in this paper, it is compared with a four-point positioning method, The principle of the four point positioning method is to select 4 points, A, B, C and D, on the measuring surface, whose coordinates and normalized magnetic source intensity value at this position are known, then the ratio relationship between the distance of the four points and the center point P of the magnetic object can be calculated using Formula (6), We set the ratio to b = BP/AP, c = CP/AP, d = DP/AP and minimize ((xP − xB)^2^ + (yP − yB)^2^ + (zP − zB)^2^ − b^2^((xP − xA)^2^ + (yP − yA)^2^ + (zP − zA)^2^)^2^) + ((xP − xC)^2^ + (yP − yC)^2^ + (zP − zC)^2^ − c^2^((xP − xA)^2^ + (yP − yA)^2^ + (zP − zA)^2^)^2^) + ((xP − xD)^2^ + (yP − yD)^2^ + (zP − zD)^2^ − d^2^((xP − xA)^2^ + (yP − yA)^2^ + (zP − zA)^2^)^2^) to obtain p-point coordinates.

An isolated cuboid model is established to compare the position estimation effects of the two methods. The method in this paper is called method 1. The four-point positioning method is called method 2. The coordinates of the center point of the red cuboid model are (0, 0, 0.35), and the length, width, and height are 0.1 m, 0.1 m, and 0.1 m. The magnetization is 1000 A/m, the magnetic inclination angle I = 90°, and the magnetic declination angle D = 0°. Two methods are used to estimate the central coordinates of the model, and the results are shown in Table 1.

It can be seen from Table 1 that the coordinates in the z direction cannot be estimated in Method 2 because the selected points are all on the same horizontal plane, and the horizontal coordinates also have large errors due to noise and other factors. The error of method 2 is small and the z-direction coordinates can be obtained.

In order to avoid the accidental error of selected points, two groups of points are added to Method 2 for calculation, and the results are shown in Table 2.

It can be seen from Table 2 that due to the influence of noise and other factors, it is difficult to obtain a stable solution using method 2 and it is not suitable for the center position estimation of magnetic objects in this paper, while the cross-correlation method is more accurate and stable.

### 2.2. Inversion Method

The planting inversion method of Uieda et al. is to set a module as the initial model (the module here is generally in the shape of a cuboid), and continue to grow based on the initial model (the model in the growth process is called the growth model), until the growth becomes the final inversion result, The growth process is constrained by the fitting residuals between the forward data of the growth model and the measured data. The specific formula [12] is as follows:(9)$$\phi_{0}(m)={\left|\left|d-p\right|\right|}_{1}=\sum_{i=1}^{N}\left|d_{i}-p_{i}\right|$$
where $d_{i}$ is the forward modeling data of the growth model of the *i*-th observation point, $p_{i}$ is the measured data of the *i*-th observation point, *N* is the number of observation points, d is a vector composed of the forward modeling data of the growth model at all observation points, and p is a vector consisting of the measured data at all observation points.

Solving the 3D structure of magnetic objects by only constraining the ***l***_1_-norm of the fitting residuals is underdetermined, and a deterministic and unique solution cannot be obtained. Therefore, additional constraints need to be introduced:(1)At least one surface of the newly added module overlaps with the existing model.

All modules that overlap at least one surface of the existing model are combined into a set, which is called the set of modules to be added. It is required that the new modules must be selected from the set of modules to be added.(2)All models are assigned the same magnetic parameters.

We assign the same size, as well as magnetization and magnetization direction to each module; however, the magnetization direction and magnetization are unknown quantities, generally determined by experience. In this paper, the normalized magnetic source strength is used for inversion, and the magnetization direction can be set as the background magnetization direction. However, the size of the magnetization will still affect the inversion results. If the magnetization is set too high, the inversion results will be too focused on the center. If the magnetization is set too low, the inversion results will be expanded and larger than the actual shape. To solve this problem, a scale factor is added to Equation (9) to obtain a new constraint function:(10)$$\phi (m)={\left|\left|\varsigma d-p\right|\right|}_{1}=\sum_{i=1}^{N}\left|\varsigma d_{i}-p_{i}\right|$$
where ς is the scale factor, which can be calculated from the forward data and the measured data. The specific formula is as follows:(11)$$\varsigma =\frac{\sum_{i=1}^{N}d_{i}p_{i}}{\sum_{i=1}^{N}d_{i}{}^{2}}$$

After adding the scale factor, the influence of the magnetization setting deviation can be weakened. The scale factor is the overall ratio of the forward data to the measured data. Multiplying the forward data by the scale factor allows the scale of the forward data to be as close to the measured data as possible. According to Equation (6), the magnitude of magnetization only affects the scale of NSS but not its distribution. Therefore, the addition of the scale factor enables each forward datum to obtain a scale close to that given by true magnetization. After the inversion results are obtained, the magnetization can be estimated by using the inversion results. We set the estimated value as the new magnetization for inversion, and repeat it many times until the estimated magnetization does not change significantly, so that more accurate magnetization estimates and inversion results can be obtained.

Constraining it only with fitted residuals may cause the shape of the inversion result to be different from the true shape. In order to improve the overall similarity of the inversion results and reduce the influence of noise, the correlation function between the forward data of the growth model and the measured data is added:(12)$$\psi (m)=1-\frac{\mathrm{cov}(d,p)}{\sqrt{D(d)D(p)}}$$
where cov (*, *) represents the covariance, and *D* (*) represents the variance.

Finally, in order to ensure the compactness of the model and prevent the model from continuing to grow in a certain direction, a space constraint function [12] is added:(13)$$\theta (m)=\frac{1}{\Delta x+\Delta y+\Delta z}\sum_{j=1}^{M}l_{j}$$
where Δx, Δy, and Δz represent the length of the growth model in the x, y, and z directions, $l_{j}$ represents the distance between each existing model and the newly added module, and *M* is the number of modules in the growth model.

Combining the above constraints, the final module selection function is:(14)Γ(m)=ϕ(m)+τψ(m)+ρθ(m)
where *m* is the magnetic parameter of the magnetic target; τ and ρ are the weight parameters, which are used to adjust the proportion between items. In each module growth process, the corresponding Γ(m) of all modules in the space to be grown is calculated, and the calculated value of each module is compared. The module with the smallest value is the optimal module in this growth process. The minimum module selection function selection equation is:(15)$$\Gamma_{\min}(m)=\min(\Gamma_{1}(m),\Gamma_{2}(m),\ldots ,\Gamma_{k}(m))$$
where k is the number of modules to be grown during this growth process.

The specific process is shown in Figure 2.

As shown on the left side of Figure 2, after the initial model is determined, the position of the module to be increased (module1,2,3,4) can be obtained, and we use the objective function to calculate each module to be increased to obtain $\Gamma_{1}(m)$, $\Gamma_{2}(m)$, $\Gamma_{3}(m)$, $\Gamma_{4}(m)$, Through comparison, the smallest Γ(m) is obtained, and its corresponding growth module with growth is the optimal growth module. After adding it to the model, the situation on the right side of Figure 2 is obtained. After the new model is obtained, the new module to be increased appears. The objective function corresponding to the new module to be increased is calculated (such as $\Gamma_{1}(m)$, $\Gamma_{2}(m)$, $\Gamma_{3}(m)$, $\Gamma_{4}(m)$, $\Gamma_{5}(m)$, $\Gamma_{6}(m)$). We compare this group of modules to obtain the optimal growth module and so on until the termination condition is reached.

After the model grows to an appropriate size, the growth needs to be terminated, and the termination condition [12] of the model growth is set as:(16)$$E(m)=\frac{\left|\phi_{new}(m)-\phi_{old}(m)\right|}{\left|\phi_{old}(m)\right|}$$
where $\phi_{old}(m)$ is the value of ϕ(m) when the optimal growth module is not added, and $\phi_{old}(m)$ is the value of ϕ(m) after adding the optimal growth module. When E(m) is less than a certain threshold, it means that the new module has little effect on the fitting, so the iteration is terminated. This threshold is generally set to 10^−4^–10^−6^. When the model has far exceeded the possible range, it proves that the threshold is set too small or the magnetic objects position parameters are calculated incorrectly. It is necessary to reset the threshold value or calculate the magnetic objects position parameters.

### 2.3. Weighting Function

In the presence of multiple magnetic sources, non-target magnetic sources often interfere with the target being inverted, causing the inversion results to shift in the horizontal position, Therefore, in the inversion process, the magnetic parameter (magnetization in this paper) needs to be weighted. We set the horizontal boundary weighting function as:(17)$$W_{h}\left(x_{i},y_{i}\right)=e^{-\frac{{\left(x_{i}-x_{0}\right)}^{2}}{L_{x}{}^{2}}-\frac{{\left(y_{i}-y_{0}\right)}^{2}}{L_{y}{}^{2}}}$$
where $x_{i}$,$y_{i}$ is the x and y coordinates of the new module, $x_{0}$,$y_{0}$ are the horizontal position coordinate value of the magnetic objects, and $L_{x}$,$L_{y}$ are the action range of the x-direction and y-direction, which can be determined according to the display range of the normalized magnetic source strength. When it is known that the interference comes from a certain direction, Equation (17) can also be rewritten as:(18)$$W_{h}\left(x_{i},y_{i}\right)=e^{\frac{{\left(-1\right)}^{q}(x_{i}-x_{0})}{L_{x}}+\frac{{\left(-1\right)}^{q}(y_{i}-y_{0})}{L_{y}}}$$
where *q* = 1 or 2. When the interference source is in the positive direction, *q* = 1, and when the interference source is in the negative direction, *q* = 2.

In this paper, in order to weaken the influence of the magnetic source remanence, the normalized magnetic source strength is used for inversion, but its vertical inversion accuracy will decrease due to its fast decay speed. In order to make up for the normalized magnetic source decay speed, we set the depth weighting function:(19)$$W_{d}\left(z_{i}\right)=e^{\beta (z_{i}{}^{2}-z_{0}{}^{2})}$$
where $z_{i}$ is the z coordinate of the new module, $z_{0}$ is the depth coordinate of the magnetic objects, and β is the focusing factor, which can be adjusted according to the thickness of the magnetic object. When the thickness of the magnetic object is unknown, β needs to be iteratively optimized by the root mean square error between the forward data of the final results and the measured data.

In the process of each growth of the model, the optimal module must be selected among the modules to be grown, and the weighting function acts in the selection process. According to the position parameter of each module in the module to be grown, the corresponding weighting function value is generated, and the weighting function value is multiplied by the forward modeling data of the corresponding module. We add the weighted forward data to the respective module selection function Γ(m), and then compare the value of the module selection function to select the optimal module. It is worth noting that the weighting function only acts on the forward process of the module to be grown. The model forward value in the module selection function Γ(m) is the sum of the weighted forward value of the module to be grown and the model forward value before growth, so the scale factor does not eliminate the effect of the weighting function.

## 3. Simulation

### 3.1. Multiple Magnetic Source Model

In order to verify the effect of the algorithm, two cuboid models as shown in Figure 3 are established for simulation. We set the ground depth to 0 m, the observation surface to −0.1 m, and the coordinate origin to be at the center of the ground. The measurement area is 3 m × 3 m, and the interval between measurement points is 0.1 m. The background magnetization direction is the magnetic inclination angle I = 90°, and the magnetic declination angle D = 0°. The coordinates of the center point of the red cuboid model are (0, 0.5, 0.9), and the length, width, and height are 0.6 m, 0.6 m, and 0.6 m. The magnetization is 140 A/m, the magnetic inclination angle I = 75°, and the magnetic declination angle D = 35°; The coordinates of the center point of the blue cuboid model are (0, −0.5, 0.7), and the length, width, and height are 0.6 m, 0.6 m, and 0.6 m. The magnetization is 100 A/m, the magnetic inclination angle I = 65°, and the magnetic declination angle D = −10°.

The model simulation data are added with Gaussian white noise 5% of the reference value to simulate the data in the actual environment. The normalized magnetic source strength and total magnetic field data of the simulation data are shown in Figure 4. Comparing the two data, it can be found that the normalized magnetic source strength has a good corresponding relationship with the actual magnetic source, but the total field data have a certain deviation from the actual magnetic source position due to the influence of the magnetization direction.

We use the magnetic source center position estimation method in Section 2.1 of this paper to estimate the position of the simulation model. The horizontal position is determined by extending the normalized magnetic source strength data of the target area upwards to find the extreme point. The range of depth estimation is set to 0–1.5 m, an equivalent source is set every 0.05 m in depth, the cross-correlation coefficients between them and the measured data are obtained, and the depth with the maximum value is taken as the estimated depth. The estimation result is shown in Figure 5, and the error between the estimation result and the actual position is shown in Table 3.

From Table 3 the estimated value is consistent with the actual value. However, this does not mean that the estimation method in this paper can achieve 100% accuracy. In fact, the accuracy of the horizontal position estimation method in this paper is determined by the interval of the measurement points, and the accuracy of the depth estimation method is determined by the interval of the equivalent sources, and a certain error will also occur when the shape of the magnetic object is irregular. The center position estimation error of the simulation results is 0, indicating that in the simulation, even if there is an error, its value is less than half of the separation distance. Such a result satisfies the accuracy required for the inversion. Therefore, the coordinates of the estimated results are used as the center coordinates of the initial model. We set the length, width, and height of each module to 0.2 m, 0.2 m, and 0.2 m, respectively, and assign the model magnetization direction as the background magnetic field direction.

In order to prove the inversion ability of the normalized magnetic source strength when the magnetic object has remanence, a control group is set up for inversion using the total magnetic field data, which is called method 2. In order to prove the role of the weighting function, a control group without adding the weighting function in the inversion process is set up, which is called method 3. The method in this paper is called method 1. The inversion results of the three methods are shown in Figure 6, Figure 7 and Figure 8. When the specific position and shape of the magnetic objects are not known, the root mean square error between the forward data and the measured data of the inversion results is often used to determine the accuracy of the inversion. Therefore, the root mean square error between the forward data of the final results of the three methods and the measured data is calculated to compare the three methods. Since the final results of the three methods are different, the estimated magnetization is also different. In order to compare the effects of the three methods more accurately, the original set’s magnetization is used for calculation. The root mean square error values are shown in Table 4.

From the inversion results in Figure 6, Figure 7 and Figure 8 and the mean square error values in Table 4 the following conclusions can be drawn: The results of method 2 have obvious offsets, and there is a large deviation from the shape position of the actual model. This is because the actual magnetization direction of the magnetic objects is rather different from the background magnetization direction. It can also be seen from the root mean square error value that the result has a larger error compared with other methods; The results of method 3 prove that the normalized magnetic source strength is not sensitive to the magnetization direction, and the inversion results do not produce a large offset. However, the results of method 2 display the phenomenon of intrusion into the non-target magnetic source, and the longitudinal boundary is not clear enough; The results of method 1 have a good correspondence with the actual model. The root mean square error value can also prove that the accuracy of the results of method 1 is much higher than that of method 2 and method 3, which proves that the method in this paper has high accuracy for the inversion of small-scale magnetic objects.

### 3.2. Isolated Magnetic Source Model

In order to verify the inversion ability of the method in this paper for irregular magnetic targets, an isolated model with step features is established. The depth range of the model is 0.4~1 m, the range of the *x*-axis direction is −0.3~0.3 m, and the range of the *y*-axis direction is −0.5~0.5 m. The specific shape is shown in Figure 9. The magnetization intensity was set to 100 A/m, the magnetization direction was set to the magnetic inclination angle I = 76°, and the magnetic declination angle D = −12°. We set the ground depth to 0 m, the observation surface to −0.1 m, and the coordinate origin to be at the center of the ground. The measurement area is 3 m × 3 m, and the interval between measurement points is 0.1 m. The background magnetization direction is the magnetic inclination angle I = 90° and the magnetic declination angle D = 0°.

The model simulation data are added with Gaussian white noise of 5% of the reference value to simulate the data in the actual environment. The normalized magnetic source strength and total magnetic field data of the simulation data are shown in Figure 10.

We use the magnetic source center position estimation method in Section 2.1 of this paper to estimate the position of the simulation model. In the depth estimation, the interval of the equivalent magnetic source is set to 0.05 m. The estimated results are shown in Figure 11.

Although the estimated result does not completely correspond to the position of the center of gravity of the model, the estimated result is within the model and can meet the inversion requirements.

The inversion is carried out according to the method in the previous section, and the obtained results are shown in Figure 12.

From the results shown in Figure 10, the deep part of the result of method 1 has parts with unclear boundaries. However, most of the features can be represented. In method 3, shallow features can be represented, but due to signal attenuation, deep features can hardly be represented. The inversion of TMI data in method 2 is seriously affected by the direction of magnetization and cannot reflect the features of magnetic objects. It can be proved that the method in this paper can better characterize the features of isolated magnetic objects.

## 4. Experiment

In order to verify the actual effect of the inversion method in this paper, a field detection experiment was carried out, and the experimental scene is shown in Figure 13. The detection device is a magnetic gradient tensor system composed of four three-axis fluxgate sensors arranged in a cross structure, and the sensors used are from Bartington in the UK. The magnetic gradient tensor value is obtained by replacing the partial differential with the difference of the magnetic field components between the two measurement points within a short distance. The magnetic gradient tensor data obtained by the difference method can effectively eliminate the influence of the Earth’s background magnetic field. Before the measurement, a series of error corrections, magnetic interference compensation, noise reduction, and other activities were carried out on the magnetic gradient tensor system to reduce the error of the measured magnetic field data [23,24]. 

The experimental process mainly includes the following steps: First, build a non-magnetic slide rail and fix the detection device on the non-magnetic slide rail; then, place the magnetic target in the field; finally, measure point by point at the same distance in the detection area, and the measurement point interval is 0.1 m. The size of the detection area is 2.1 m × 2.1 m. The southwest corner of the non-magnetic slide rail is used as the coordinate origin, the east direction is the *x*-axis, the north direction is the *y*-axis, and the vertical downward direction is the *z*-axis to establish a plane rectangular coordinate system. The coordinates of the center point of the cylinder are (0.693 m, 1.499 m, and 0.155 m), the diameter is 0.106 m, and the length is 0.464 m. The coordinates of the center point of the cuboid are (1.512 m, 0.749 m, and 0.152 m), the length in the north-south direction is 0.358 m, the length in the east-west direction is 0.234 m, and the thickness is 0.1 m. The background magnetization direction is the magnetic inclination angle I = 56° and the magnetic declination angle D = −16°. Two magnetic objects are set at similar heights to simulate the interference of non-target magnetic sources underground.

The data collected from the experiment are shown in Figure 14. Using the method in this paper to estimate the center position of magnetic objects, the depth estimation range is set to 0–0.6 m, and the interval of the equivalent source is set to 0.01 m. The results are shown in Figure 15. The estimated center position results and errors are shown in Table 5.

From the results in Figure 15 and Table 5 the errors of the estimated results are all within the theoretical error; that is, the error results are not greater than half of the measurement interval and the equivalent source setting interval. It is proved that the center position estimation method in this paper is effective, but limited by the size of the measurement interval.

The initial model is set up with the estimated center position. We set the length, width, and height of each module to 0.1 m, 0.1 m, and 0.1 m, respectively, and assign the model magnetization direction as the background magnetic field direction, and then the magnetic target is inverted. A control group was set up for inversion using the total magnetic field data, called method 2, while the method in this paper is called method 1, and the results are shown in Figure 16. The RMSE values of the forward data and the measured data of the final inversion results are shown in Table 6.

From the inversion results in Figure 16 and the root mean square error values in Table 6 in the real magnetic field environment, the method in this paper can still obtain relatively accurate 3D inversion results. In practical situations, the estimation of the magnetization direction of magnetic objects in a complex magnetic field environment is avoided, and a high inversion accuracy can be maintained.

## 5. Conclusions

Focusing on the problems existing in the process of 3D inversion of small-scale magnetic targets, this paper proposes a method for 3D inversion using normalized magnetic source strength. We determined the position of the initial seed by estimating the position of the magnetic source before inversion and the normalized magnetic source strength was used for inversion, which reduces the influence of magnetic remanence. The original planting inversion method was improved, so that the inversion results are more similar to the actual magnetic properties. In the inversion process, a weighting function is added to reduce the precision drop caused by the interference of non-target magnetic sources and the rapid attenuation of the own signal.

The simulation and experimental results show that this method is less affected by the magnetization direction of the magnetic object, and has high precision even in the presence of non-target magnetic sources and noise interference. In the actual detection process of small-scale magnetic objects, the situation of the magnetic source is often complicated. The residual magnetism of magnetic objects and the interference of non-target magnetic sources will cause large errors in the inversion results. The purpose of this paper is to provide theoretical support for the inversion of small-scale magnetic objects in such practical situations.

## Figures and Tables

**Figure 1 micromachines-13-01813-f001:**
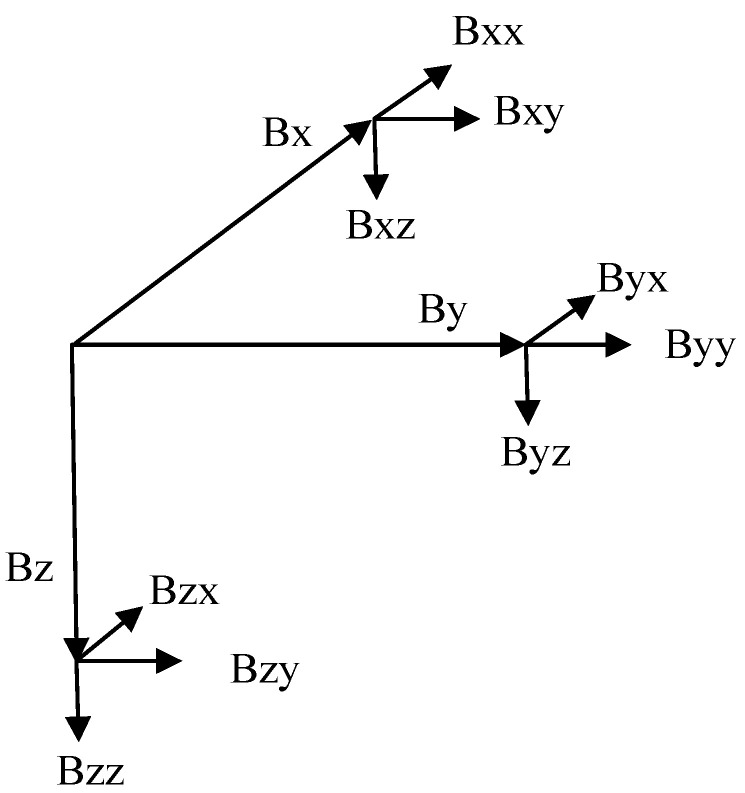
Schematic diagram of magnetic field component and magnetic gradient tensor.

**Figure 2 micromachines-13-01813-f002:**
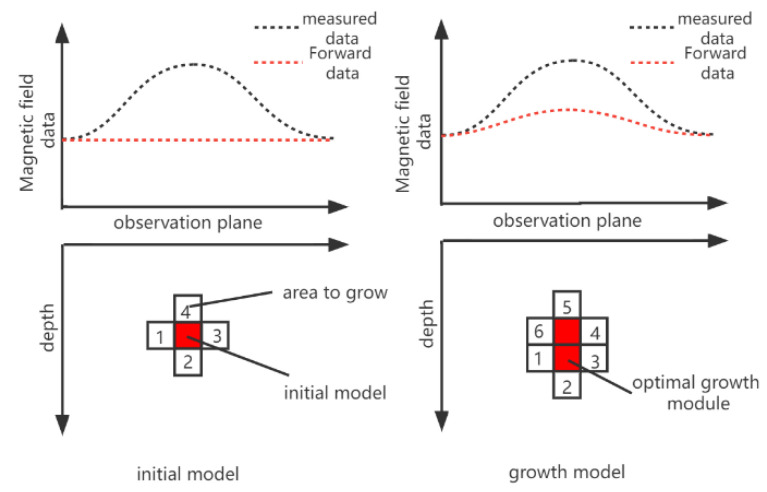
Module growth process.

**Figure 3 micromachines-13-01813-f003:**
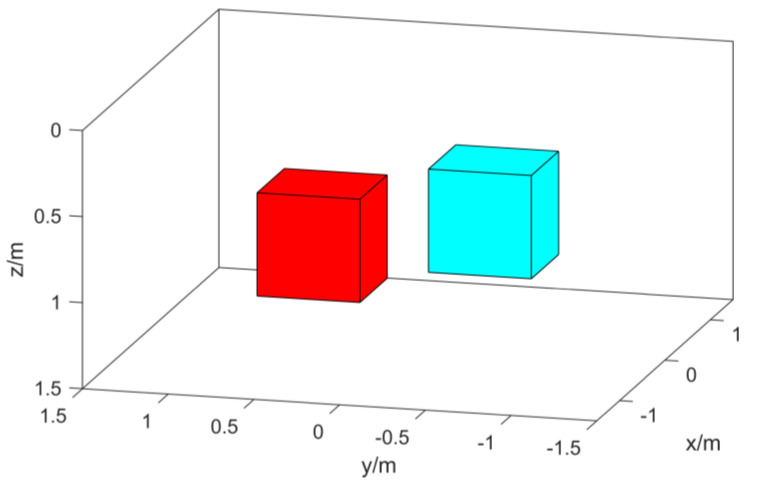
Schematic diagram of the simulation model.

**Figure 4 micromachines-13-01813-f004:**
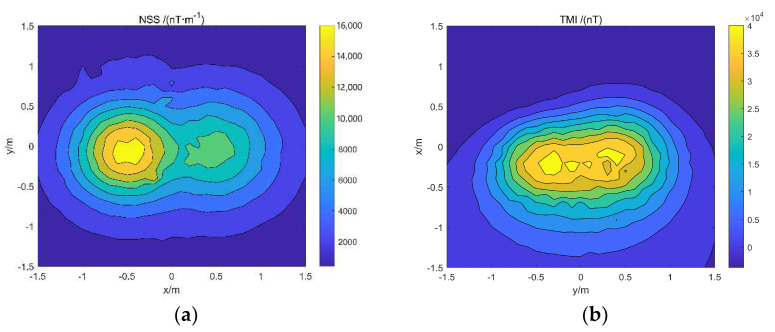
Model simulation data: (**a**) Simulation data for normalized magnetic source strength; (**b**) simulation data of the total magnetic field.

**Figure 5 micromachines-13-01813-f005:**
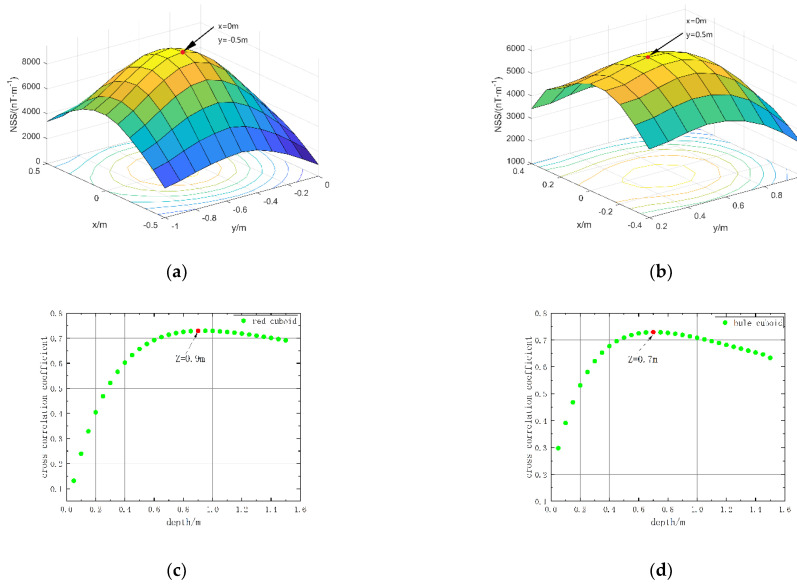
Estimated results of the center position of the simulation model: (**a**) Horizontal position of the red cuboid; (**b**) horizontal position of the blue cuboid; (**c**) the red cuboid depth; (**d**) the blue cuboid depth.

**Figure 6 micromachines-13-01813-f006:**
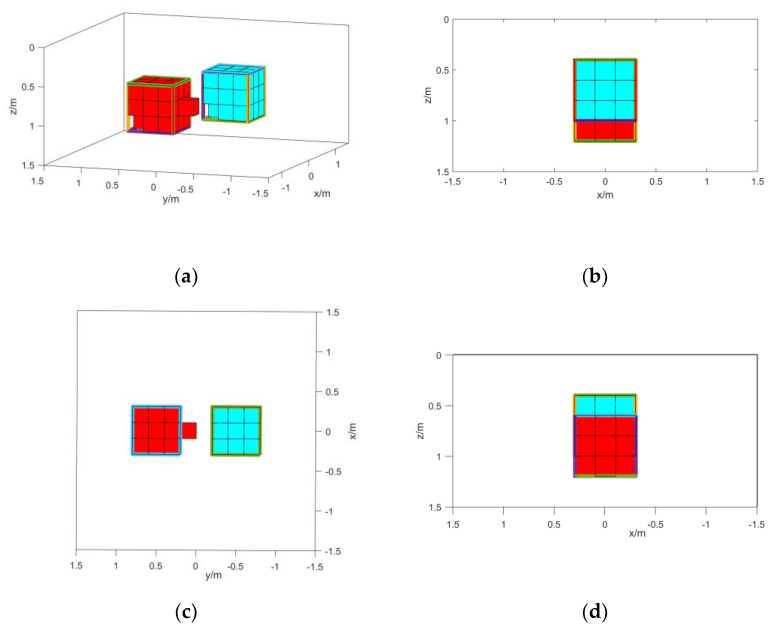
(**a**) Method 1 inversion results; (**b**) right-side view of method 1 inversion results; (**c**) top view of method 1 inversion results; (**d**) left-side view of method 1 inversion results.

**Figure 7 micromachines-13-01813-f007:**
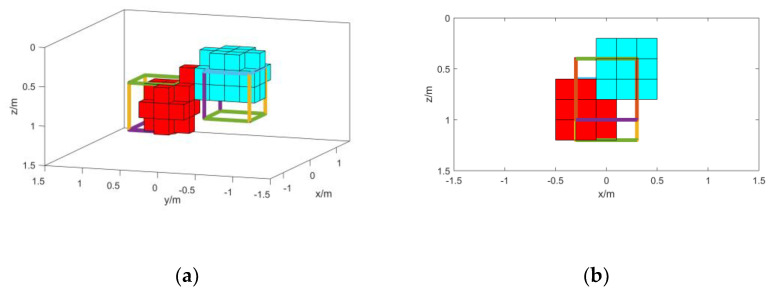
(**a**) Method 2 inversion results; (**b**) right-side view of method 2 inversion results; (**c**) top view of method 2 inversion results; (**d**) left-side view of method 2 inversion results.

**Figure 8 micromachines-13-01813-f008:**
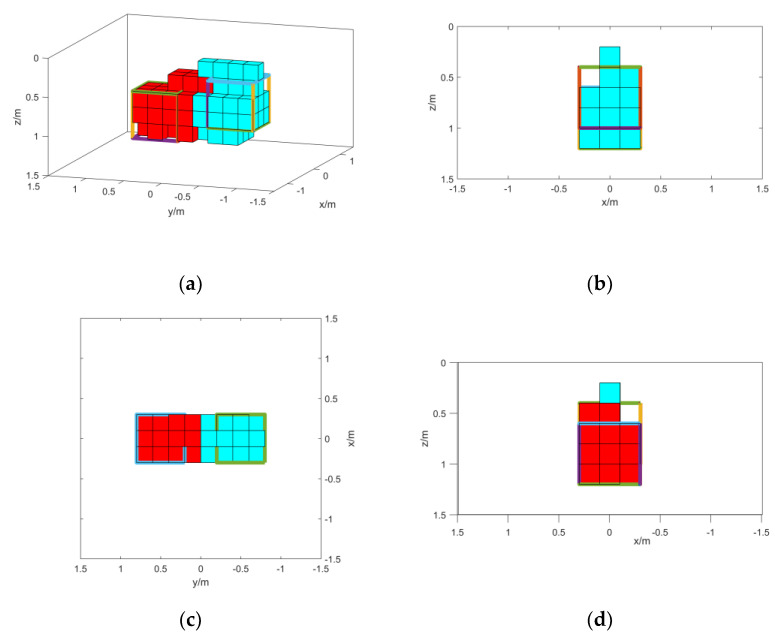
(**a**) Method 3 inversion results; (**b**) right-side view of method 3 inversion results; (**c**) top view of method 3 inversion results; (**d**) left-side view of method 3 inversion results.

**Figure 9 micromachines-13-01813-f009:**
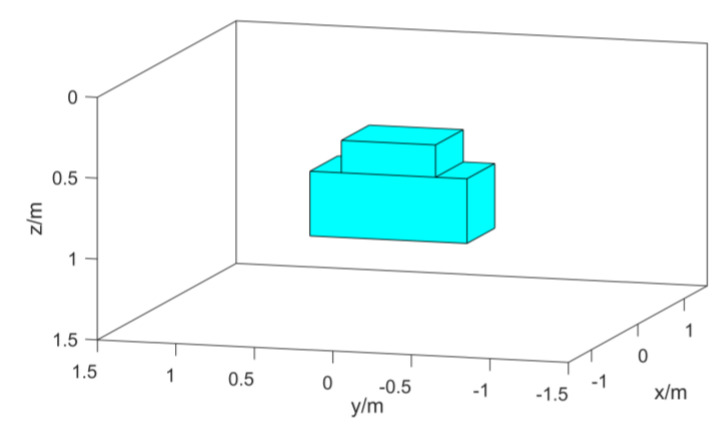
Schematic diagram of the isolated model.

**Figure 10 micromachines-13-01813-f010:**
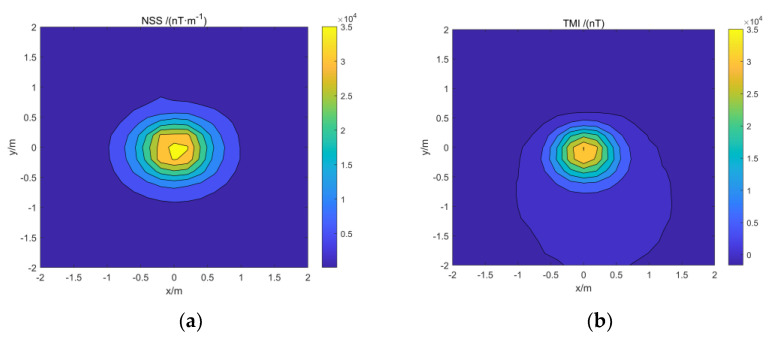
Isolated model simulation data: (**a**) Simulation data for normalized magnetic source strength; (**b**) simulation data of the total magnetic field.

**Figure 11 micromachines-13-01813-f011:**
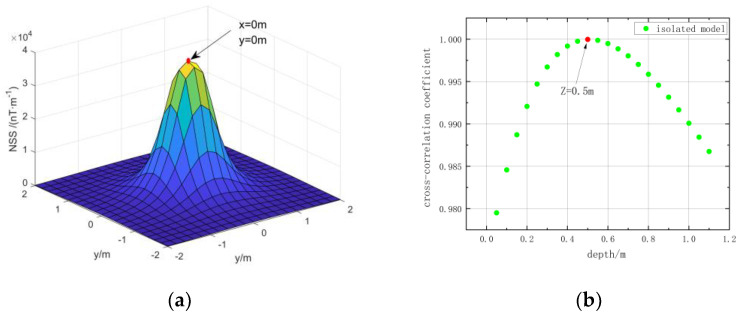
Estimated results of the center position of the simulation model; (**a**) Horizontal position of the model; (**b**) horizontal position of the model.

**Figure 12 micromachines-13-01813-f012:**
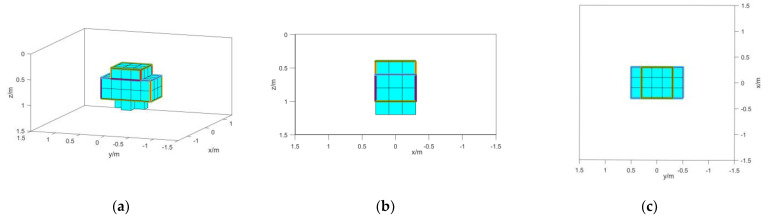
(**a**) Method 1 inversion results; (b) left- side view of method 1 inversion results; (c) top view of method 1 inversion results; (d) method 2 inversion results; (e) left- side view of method 2 inversion results; (f) top view of method 2 inversion results; (g) method 3 inversion results; (h) left- side view of method 3 inversion results; (i) top view of method 3 inversion results.

**Figure 13 micromachines-13-01813-f013:**
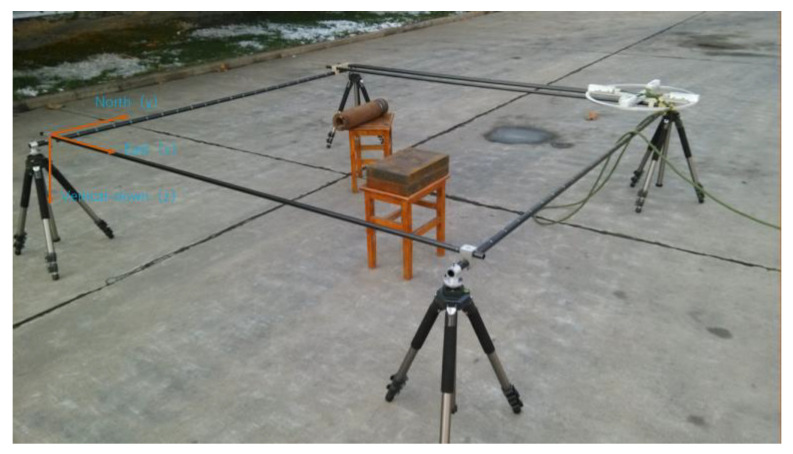
Experimental scene.

**Figure 14 micromachines-13-01813-f014:**
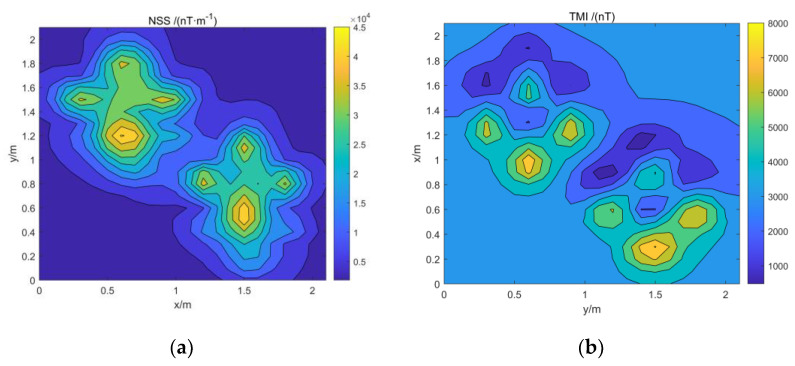
Actual data: (**a**) Actual data for normalized magnetic source strength; (**b**) actual data of the total magnetic field.

**Figure 15 micromachines-13-01813-f015:**
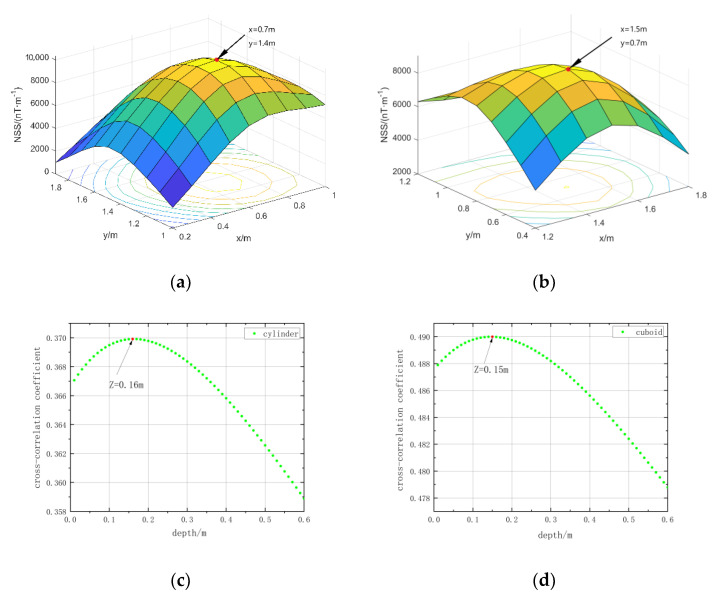
Estimated results of the center position of the measured magnetic object: (**a**) Horizontal position of cylinder; (**b**) horizontal position of cuboid; (**c**) cylinder depth; (**d**) cuboid depth.

**Figure 16 micromachines-13-01813-f016:**
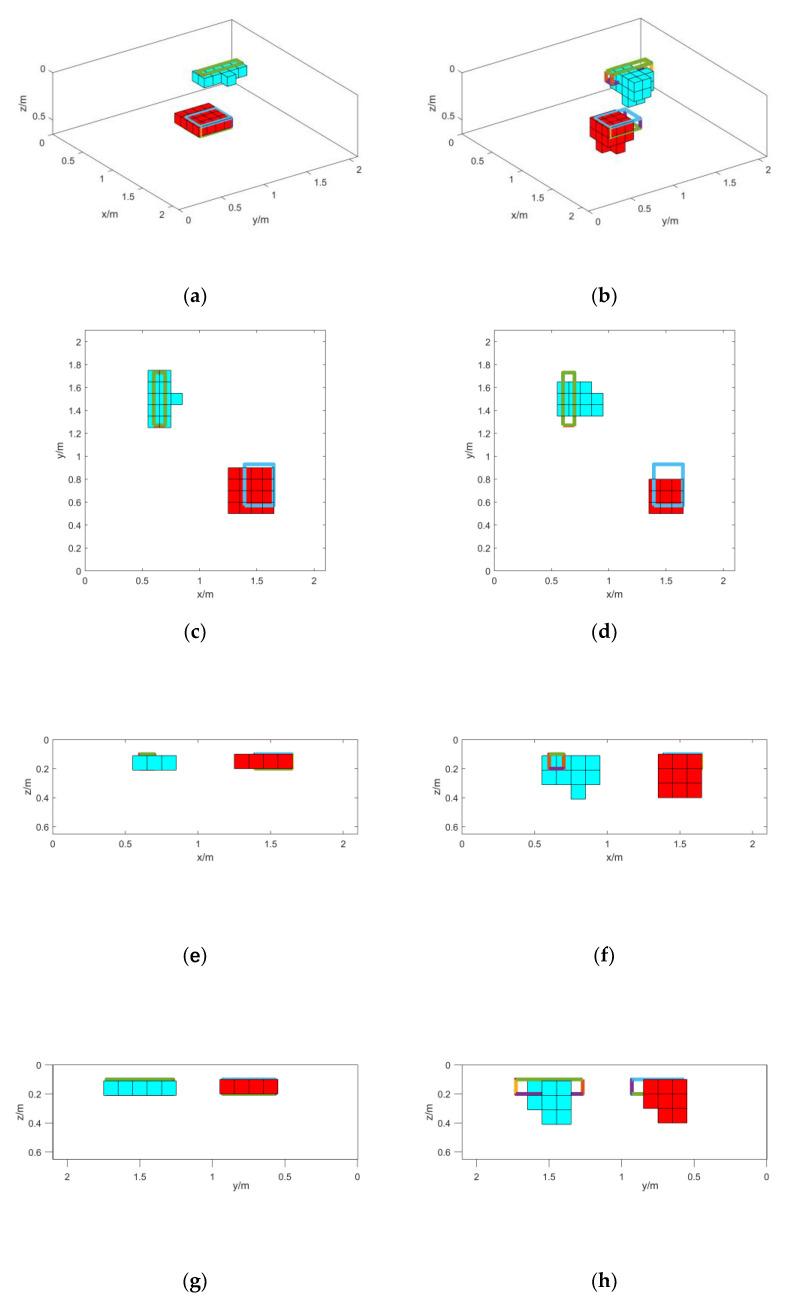
Inversion results: (**a**) Method 1 inversion results; (**b**) method 2 inversion results; (**c**) top view of method 1 inversion results; (**d**) top view of method 2 inversion results. (**e**) main view of method 1 inversion results; (**f**) main view of method 2 inversion results; (**g**) left view of method 1 inversion results; (**h**) left view of method 2 inversion results.

**Table 1 micromachines-13-01813-t001:** Central coordinate estimation results of two methods.

	X(m)	Y(m)	Z(m)
Method 1	0	0	0.36
Method 2	0.89	0.89	-

**Table 2 micromachines-13-01813-t002:** Results of method 2 in different groups of points.

	X(m)	Y(m)	Z(m)
Group 1	0.71	0.71	-
Group 2	0.89	0.89	-
Group 3	1	1	-

**Table 3 micromachines-13-01813-t003:** The center position estimation result and absolute error.

	Red Cuboid Model			Blue Cuboid Model		
	X(m)	Y(m)	Z(m)	X(m)	Y(m)	Z(m)
estimated value	0	0.5	0.9	0	−0.5	0.7
error	0	0	0	0	0	0

**Table 4 micromachines-13-01813-t004:** Root mean square error between inversion results and measured data under different methods.

	Method 1	Method 2	Method 3
Normalized Magnetic Source Strength (nT/m)	92.247	3161.43	1413.336

**Table 5 micromachines-13-01813-t005:** The center position estimation result and absolute error.

	Cylinder			Cuboid		
	X(m)	Y(m)	Z(m)	X(m)	Y(m)	Z(m)
estimated value	0.7	1.4	0.16	1.5	0.7	0.15
error	0.007	−0.099	0.005	−0.012	−0.049	−0.002

**Table 6 micromachines-13-01813-t006:** Root mean square error between inversion results and measured data under different methods.

	Method 1	Method 2
Normalized Magnetic Source Strength (10^3^ nT/m)	105.709	131.197

## Data Availability

Not applicable.

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
