# Peer review of "A Three-Dimensional Inversion Method for Small-Scale Magnetic Objects Based on Normalized Magnetic Source Strength"

_micromachines, 2022, doi:10.3390/mi13111813_

Round 1
Reviewer 1 Report
Comment on micromachines-1939406-peer-review-v1
The paper presents a method to localize objects made of magnetic material through their stray field. Is this method more efficient than that of usual pipe locating equipment? Moreover, usual pipe locating equipment is able to localize non ferromagnetic metals.
There is a typo in equation (3). The exponent of the second lambda is 1 not 2.
In equation (4), the exponent is not |n| but 4.
According to equation (4), the variation of NSS with respect to z is described by the law µ/(x²+y²+z²)². Why use equation (5) ?
With measurements at 4 points A,B,C,D, the position of the source P can be determined: from µ/(AP)⁴, µ/(BP)⁴, µ/(CP)⁴, µ/(DP)⁴, the 3 ratios b = BP/AP, c = CP/AP, d = DP/AP are evaluated; these 3 ratios give enough information for determination of the position of P; the 3 coordinates of P are deduced by minimizing
( (xP -xB)² + (yP -yB)² + (zP -zB)² - b² ((xP -xA)² + (yP -yA)² + (zP -zA)²)² +
( (xP -xC)² + (yP -yC)² + (zP -zC)² - c² ((xP -xA)² + (yP -yA)² + (zP -zA)²)² +
( (xP -xD)² + (yP -yD)² + (zP -zD)² - d² ((xP -xA)² + (yP -yA)² + (zP -zA)²)²
Why use equation (6) ?
In equation (14), are there Lx, Ly or Lx², Ly² ?
In line 241, is there “equation (13)” or “equation (14)” ?
In equation (16), is there “? zi² -z0²” or “? (zi² -z0²)” ?
The experiment is carried out in the air. But earth may contain iron oxide having magnetic properties. What would be the efficiency of the method to detect buried objects?
Author Response
Dear Reviewer:
Thank you for your approval of this work of our team with precious letter and for the comments concerning our manuscript entitled “A three-dimensional inversion method for small-scale magnetic objects based on normalized magnetic source strength” (ID: micromachines-1939406). Those comments are all valuable and very helpful for revising and improving our paper, as well as the important guiding significance to our researches. We have studied comments carefully and have made correction which we hope meet with approval. Revised portion are marked in yellow in the paper.
Here are our responses to your review comments.
Point 1: The paper presents a method to localize objects made of magnetic material through their stray field. Is this method more efficient than that of usual pipe locating equipment? Moreover, usual pipe locating equipment is able to localize non ferromagnetic metals.
Response 1: I'm sorry that we did not clarify the scope of use when describing the method in this article. The method in this paper uses magnetic field data to detect magnetic objects, so this method cannot detect non-magnetic objects. This means that this method is not suitable for all pipe detection, but it does not mean that this method is useless for ferromagnetic targets, especially unexploded ordnance, the method in this paper can obtain its specific shape to guide the subsequent elimination work and reduce the danger of unexploded ordnance elimination. We have revised the Introduction section to describe in detail the scope of applicability of our method, e.g., " In the process of detecting some important or dangerous small magnetic targets (such as unexploded ordnance, Ferromagnetic underground pipes, etc.), We hope to obtain the shape and position of the target, so as to guide the follow-up unexploded ordnance removal and other work, and reduce the danger or loss caused by the removal of unexploded ordnance, etc. Therefore, it is necessary to perform 3D inversion of these magnetic targets by using the magnetic field data ". Hope our explanation and handling clear your doubts, thanks again for your professional comment!
Point 2: There is a typo in equation (3). The exponent of the second lambda is 1 not 2.
Response 2: I am very sorry for the typo, the corresponding formula has been corrected, thank you very much for the correction
Point 3: In equation (4), the exponent is not |n| but 4.
Response 3: Thanks for your detailed guidance, the |n| in the formula has been changed to 4. Thank you very much for your reminder.
Point 4: According to equation (4), the variation of NSS with respect to z is described by the law µ/(x²+y²+z²)². Why use equation (5) ?
Response 4: We are very sorry that equation (5) and related descriptions in the article may be ambiguous. Actually equation (5) does not conflict with equation (4), equation (4) is the forward formula for the known magnetic source, and equation (5) is the frequency domain conversion formula when the data of a certain measurement surface is known. In the actual measurement process, we often can only obtain the data of a certain measurement surface without knowing the magnetic source (such as the magnetization and position of the magnetic source, etc.) So, it is not possible to use equation (4) to get different height data. This means that we cannot obtain other data by changing z in µ/(x²+y²+z²)² during the actual detection process, so we add the frequency domain conversion formula equation (5) that can obtain other surface data by measuring surface data. We supplement the function of this formula in the corresponding part of the text e.g.” In the actual measurement process, we can only obtain the measurement data of a certain height, so we use the method of frequency domain conversion, and use the Fourier transform to obtain the normalized magnetic source at different heights.”. Thank you for your detailed comments that make this article more logical and structured.
Point 5: With measurements at 4 points A,B,C,D, the position of the source P can be determined: from µ/(AP)⁴, µ/(BP)⁴, µ/(CP)⁴, µ/(DP)⁴, the 3 ratios b = BP/AP, c = CP/AP, d = DP/AP are evaluated; these 3 ratios give enough information for determination of the position of P; the 3 coordinates of P are deduced by minimizing
( (xP -xB)² + (yP -yB)² + (zP -zB)² - b² ((xP -xA)² + (yP -yA)² + (zP -zA)²)² +
( (xP -xC)² + (yP -yC)² + (zP -zC)² - c² ((xP -xA)² + (yP -yA)² + (zP -zA)²)² +
( (xP -xD)² + (yP -yD)² + (zP -zD)² - d² ((xP -xA)² + (yP -yA)² + (zP -zA)²)²
Why use equation (6) ?
Response 5: We admire your erudition and expertise and thank you for your new methods and ideas. The method you propose can get more precise location coordinates. However, since this method selects the data of four separate points, and the data of each point has an influence on the result, the calculation result will be easily disturbed by noise in the actual measurement process. Therefore, it is necessary to select the data of multiple sets of points and add noise processing methods. How to choose the optimal data, which noise reduction method to use, and the subsequent verification process require more space, so we hope to further study the method you proposed in the follow-up work. Although the resolution of the method equation (6) in this paper is not as high as that of your proposed method, it has better stability in a noisy environment, and the accuracy of the obtained results can meet the subsequent inversion requirements. So, we decided to use the original method. We added supplementary notes e.g. " Equivalent source cross-correlation method makes full use of the advantages of grid data and applies enough data to the localization process. And the cross-correlation method has good tolerance to noise, so this method can obtain better depth estimation results in the actual measurement environment with noise." to the description of the original method. I hope we can get your understanding for our choice, and thank you again for your help.
Point 6: In equation (14), are there Lx, Ly or Lx², Ly² ?
Response 6: Thank you for your careful and professional guidance, Formula (14) should be Lx², Ly², we have changed Lx, Ly in the original formula to Lx², Ly².
Point 7: In line 241, is there “equation (13)” or “equation (14)” ?
Response 7: I'm very sorry for the typo, it should be equation (14), We have corrected this part. Thank you for your reminder.
Point 8: In equation (16), is there “? zi² -z0²” or “? (zi² -z0²)” ?
Response 8: Thank you for your careful inspection, it should be ? (zi² -z0²), The "? zi² -z0²" in the formula has been changed to "? (zi² -z0²)", thank you very much for your reminder.
Point 9: The experiment is carried out in the air. But earth may contain iron oxide having magnetic properties. What would be the efficiency of the method to detect buried objects?
Response 9: Thank you for your valuable comments on the experiment,The magnetic iron oxide contained in the Earth you mentioned does have an effect on the inversion,This situation is discussed in the following categories: (1) The magnetic field generated by the tiny magnetic iron oxide with weak magnetism is generally contained in the background magnetic field of the earth。During the detection process, we use the difference method to obtain the magnetic gradient tensor data, which can eliminate most of the influence of the earth's background magnetic field. And the research group has done a lot of research on magnetic gradient tensor system error correction, magnetic interference compensation, and noise reduction in the previous work. And it has been proved that it can greatly weaken the influence of magnetic interference and various errors in the environment [1][2]. In order to allow readers to better understand this process, we have added a narrative " The magnetic gradient tensor data obtained by the difference method can effectively eliminate the influence of the Earth's background magnetic field. Before the measurement, a series of error correction, magnetic interference compensation, noise reduction and other work were carried out on the magnetic gradient tensor system to reduce the error of the measured magnetic field data." to the text. So far, we believe that with the aid of system calibration and magnetic interference compensation, the method in this paper will have certain applicability in various interference environments whether underground or in the air;(2) For magnetic iron oxide interference with a volume comparable to the target body, but at a certain distance from the target body. Such magnetic iron oxides will be inverted as non-targets. In this paper, two magnetic bodies are set up in the experiment to simulate the inversion situation with non-target interference. According to the experimental results, for the interference of non-target objects, the inversion results still have high accuracy. During the experiment, we placed the two magnetic bodies at similar heights, which could simulate the interference of non-target bodies under the ground. We have added relevant supplementary notes " Two magnetic objects are set at similar heights to simulate the interference of non-target magnetic sources underground." in the text. (3) For the interference of magnetic iron oxide with a volume equivalent to the target body and almost coincident with the target body. In this case, the magnetic sources cannot be distinguished, and the inversion method will be used as the same target for inversion. Only by the three-dimensional shape of the inversion result can distinguish the target body from the non-target body. This kind of case is the same as case (2), no matter in the ground or in the air, the same inversion result will be obtained. Combining the above three situations, although the experiments in this paper are carried out in the air, the results are the same as those placed under the ground. Therefore, we believe that the method in this paper can detect underground objects.
I hope our explanation can satisfy you, and thank you for your professional opinion. Your comments provide a powerful reference for our follow-up experimental design, and we look forward to having more discussions and exchanges with you in the future.
The following are related references:
[1] Q. Li, Z. Shi, Z. Li, H. Fan, and G. Zhang, “Preferred Configuration and Detection Limits Estimation of Magnetic Gradient Tensor System,” in IEEE Transactions on Instrumentation and Measurement, vol. 70, pp. 1-14, Oct. 2021, Art no. 1010214.
[2] Q. Li, Z. Li, Y. Zhang, H. Fan, and G. Yin, “Integrated Compensation and Rotation Alignment for Three-Axis Magnetic Sensors Array,” in IEEE Transactions on Magnetics, vol. 54, no 10, pp. 1-11, Oct. 2018.

Reviewer 2 Report
Dear Authors
The paper is sounding and improved the 3D inversion process by implementing the NSS’s properties, the unique maximum point, and the upward continuation of NSS to estimate the depth using the cross-correlation of the equivalent sources.
However, I have some minor comments as follows:
1- The introduction section seems voluminous, broad, and heterogeneous. The authors are supposed to focus on the main topic of the study and present a complete Literature Review to make research gaps and innovations easy to detect. Authoritative synthesis assessing the current state-of-the-art is absent.
2- The current introduction is very simple and misses many contents related to the problem formulation. There is not a clear categorization of related works.
3- I suggested some up-to-date references dealing with this subject. Please revise the attached PDF
Regards

Author Response
Dear Reviewer:
Thank you for your approval of this work of our team with precious letter and for the comments concerning our manuscript entitled “A three-dimensional inversion method for small-scale magnetic objects based on normalized magnetic source strength” (ID: micromachines-1939406). Those comments are all valuable and very helpful for revising and improving our paper, as well as the important guiding significance to our researches. We have studied comments carefully and have made correction which we hope meet with approval. Revised portion are marked in yellow in the paper.
Here are our responses to your review comments.
Point 1: The introduction section seems voluminous, broad, and heterogeneous. The authors are supposed to focus on the main topic of the study and present a complete Literature Review to make research gaps and innovations easy to detect. Authoritative synthesis assessing the current state-of-the-art is absent.
Response 1: Thank you for your professional opinion, which will greatly help the quality and readability of our paper. You point out that the main topic of the study in the introduction is not sufficiently prominent. We think this problem is very serious, so according to your suggestion, We remove redundant descriptions and analyze and compare various inversion methods, highlighting the strengths and weaknesses of existing methods. The specific modifications are marked in yellow in the text. We hope our changes meet your expectations and look forward to more discussions and exchanges with you in the future.
Point 2: The current introduction is very simple and misses many contents related to the problem formulation. There is not a clear categorization of related works.
Response 2: We admire your professionalism and meticulousness, and thank you for pointing out the problem for us. We will describe the current method in more detail and organize and classify the original content. The specific modifications are marked in yellow in the text. We hope our processing can satisfy you, thank you again for your professional review.
Point 3: I suggested some up-to-date references dealing with this subject. Please revise the attached PDF
Response 3: Thank you for your support and help. The references you provide are necessary to supplement the background of the article, and we have added them to the references of this article. Thanks again for your guidance.

Reviewer 3 Report
Dear Editor
The manuscript explains an inversion approach for NSS data. It is novel and prepared well. However, I have some comments that should be regarded for acceptance of the manuscript as follows:
1- Most formulas have no reference
2- In Eq. 3, the authors must explain i as Beiki et al. (2012)
3- How do authors set magnetization bound in the inversion process? How does the scale factor deal with this issue? Please explain more.
4- The inversion algorithm that minimizes the objective function (Eq. 12) must be added to the manuscript.
5- How do the authors incorporate weighting matrices in the inversion process?
6- another synthetic example is required
7- please add cross sections in the models to see changes in magnetization.
Author Response
Dear Reviewer:
Thank you for your approval of this work of our team with precious letter and for the comments concerning our manuscript entitled “A three-dimensional inversion method for small-scale magnetic objects based on normalized magnetic source strength” (ID: micromachines-1939406). Those comments are all valuable and very helpful for revising and improving our paper, as well as the important guiding significance to our researches. We have studied comments carefully and have made correction which we hope meet with approval. Revised portion are marked in yellow in the paper.
Here are our responses to your review comments.
Point 1: Most formulas have no reference
Response 1: Thank you for your careful review of the manuscript, and we have followed your request to annotate the sources of the relevant formulas. Formulas (1) and (2) are conventional matrix transformations; formulas (3), (4) and (5) are derived from the papers published by Pedersen et al in 1990(Pedersen, L. B., & Rasmussen, T. M. (1990).The gradient tensor of potential field anomalies: Some implications on data collection and data processing of maps. Geophysics, 55(12), pp:1558–1566.);Equation (6) is derived from the paper published by Beiki et al in 2012(Beiki M., Clark D.A., Austin J.R., Foss C.A., Estimating source location using normal-ized magnetic source strength calculated from magnetic gradient tensor data. Geophysics.2012,77, pp: 23–37. ); Formulas (9), (13), (16) are derived from the paper published by Uieda et al. in 2012(Uieda L, Barbosa VCF. Use of the"shape-of-ano-maly" data misfit in 3D inversion by planting anomalous densities[C]. SEG Technical Program Expanded Abstracts, 2012, 31, pp:3160-3164.). All other formulas are original formulas in this article. Hope our modification meets your requirements.
Point 2: In Eq. 3, the authors must explain i as Beiki et al. (2012)
Response 2: Thank you for your reminder. i in Equation 3 represents different eigenvalues, namely λ1, λ2, λ3, We will add the relevant explanation to the corresponding position. And in order to make readers understand better, we explain the derivation formula of each eigenvalue according to the relevant references. Thank you for your detailed review.
Point 3: How do authors set magnetization bound in the inversion process? How does the scale factor deal with this issue? Please explain more.
Response 3: We are very sorry that we did not state this clearly in the manuscript, we will explain this further. This paper adopts an inversion method based on the growth model to fit the shape of the magnetic body. Therefore, before inversion, we need to set a value of magnetization such as 100 A/m, so that each module is assigned this value. This means that in the inversion space where there are modules, the magnetization is set to 100 A/m, and where there are no modules, the magnetization is 0 A/m. The boundary of the model is the boundary between the magnetization of 0 and the set value. This magnetization set value is generally estimated from the measured magnetic field data values in conjunction with the approximate volume of the magnetic target. Such empirical setting forces the choice of magnetization to be within a larger range. Therefore, in order to avoid this situation to bring errors to the inversion, we designed a scale factor. The scale factor is the overall ratio of the forward data to the measured data. Multiplying the forward data by the scale factor allows the scale of the forward data to be as close to the measured data as possible. According to the forward process of NSS, the magnitude of magnetization only affects the scale of NSS but not its distribution. Therefore, the addition of the scale factor enables each forward data to obtain a scale close to that given by the true magnetization. We will add relevant explanations to the manuscript and highlight them in yellow.
Hope our explanation and handling clear your doubts, thanks again for your professional comment!
Point 4: The inversion algorithm that minimizes the objective function (Eq. 12) must be added to the manuscript.
Response 4: Thank you for your professional opinion, we really did not fully explain in this part of the manuscript. So following your comments, we added the selection function for the optimal growth module, the minimization objective function you mentioned. Since this paper adopts an inversion method based on the growth model to fit the shape of the magnetic body, in the process of each model growth, it is necessary to select the optimal growth module among the modules to be added. Therefore, (Equation 12) may be more aptly called the module selection function than the objective function. Its main function is to calculate the effect of the module on the overall fit after each corresponding module in the module to be added is added to the model. Therefore, each time the model grows, the number of modules to be grown is equal to the number of equations to be calculated. By comparing the function values corresponding to each module to be grown calculated in this growth process, the optimal module can be selected. Therefore, we have incorporated the formula for comparing the magnitude of the function values as the selection function for the optimal growth module into the manuscript. Hope our processing meets your requirements.
Point 5: How do the authors incorporate weighting matrices in the inversion process?
Response 5: Thank you for your review comments on the weighting function, reminding us that the relevant detailed description is missing from the manuscript. We will provide additional explanations for this. In the process of each growth of the model, the optimal module must be selected among the modules to be grown, and the weighting function acts in the selection process. According to the position parameter of each module in the module to be grown, the corresponding weighting function value is generated, and the weighting function value is multiplied by the forward modeling data of the corresponding module. Add the weighted forward data to the respective module selection function , and then compare the value of the module selection function to select the optimal module. It is worth noting that the weighting function only acts on the forward process of the module to be grown. The model forward value in the module selection function is the sum of the weighted forward value of the module to be grown and the model forward value before growth, so the scale factor does not eliminate the effect of the weighting function. Hope our explanation clears your doubts, and thanks again for your comments on the manuscript.
Point 6: another synthetic example is required
Response 6: Thank you for your constructive comments on our manuscript, we speculate that you might want us to add models of different shapes for simulation verification. Therefore, we added an isolated model with stepped features to verify the inversion method, The relevant simulation results are added to the manuscript and highlighted in yellow(on pages 12-15). Hope our work meets your expectations.
Point 7: another synthetic example is required
Response 7: Thank you for your professional advice on the content of the manuscript. A single view really doesn't show the results of the inversion better. Combined with your suggestion, we will show the inversion results of the model more fully. However, considering that the boundary of the inversion results obtained by the method in this paper is the boundary between the magnetization of 0 and the set value, that is, the inner magnetization of the model is equal everywhere, and the outer magnetization of the model is 0. And according to the characteristics of the inversion method, there is no cavity with 0 magnetization inside the inversion result. So the way to increase the cross section is not as intuitive as adding other views. Therefore, we adopt the multi-view display of the inversion results, which makes the inversion results more clearly displayed. Thank you for your suggestion, and we hope that the approach we have adopted can gain your approval.

Round 2
Reviewer 1 Report
the revised version is suitable for publication
Author Response
Thank you very much for your recognition and look forward to further exchanges and discussions with you in the future.